# Estrogen and Progesterone Receptors Are Dysregulated at the BPH/5 Mouse Preeclamptic-Like Maternal–Fetal Interface

**DOI:** 10.3390/biology13030192

**Published:** 2024-03-16

**Authors:** Viviane C. L. Gomes, Bryce M. Gilbert, Carolina Bernal, Kassandra R. Crissman, Jenny L. Sones

**Affiliations:** 1Department of Small Animal Clinical Sciences, Michigan State University College of Veterinary Medicine, East Lansing, MI 48824, USA; leitegom@msu.edu (V.C.L.G.);; 2Department of Veterinary Clinical Sciences, Louisiana State University School of Veterinary Medicine, Baton Rouge, LA 70803, USA; 3Equine Reproduction Laboratory, Colorado State University, Fort Collins, CO 80521, USA

**Keywords:** preeclampsia, decidua, stroma, endothelium, sex hormone receptor

## Abstract

**Simple Summary:**

Preexisting reproductive disorders and perturbed placentation are proposed to play a role in the development of preeclampsia, a leading hypertensive disorder of pregnancy. However, underlying mechanisms remain incompletely understood. Pregnancy establishment and maintenance are tightly regulated by estrogen and progesterone signaling in the uterus and developing placenta. Accordingly, placental estrogen and progesterone receptor dysregulations have been speculated to contribute to preeclampsia. Using a spontaneous model of superimposed preeclampsia, the Blood Pressure High Subline 5 (BPH/5) mouse, we tested the hypothesis that uteroplacental estrogen and progesterone receptor misexpression occur prior to pregnancy and during the peri-implantation period of preeclamptic-like pregnancies. BPH/5 females display estrogen deficiency, delayed embryonic development, and delayed decidualization. Herein, we describe for the first time estrogen and progesterone receptor dysregulation in the BPH/5 non-pregnant uterus and developing maternal–fetal interface. This study provides evidence of disrupted sex hormone signaling in the peri-conception phase of preeclamptic-like BPH/5 pregnancies, offering potential insights on estrogen and progesterone signaling at unexplored timepoints of human preeclampsia.

**Abstract:**

The etiopathogenesis of preeclampsia, a leading hypertensive disorder of pregnancy, has been proposed to involve an abnormal circulating sex hormone profile and misexpression of placental estrogen and progesterone receptors (ER and PR, respectively). However, existing research is vastly confined to third trimester preeclamptic placentas. Consequently, the placental–uterine molecular crosstalk and the dynamic ER and PR expression pattern in the peri-conception period remain overlooked. Herein, our goal was to use the BPH/5 mouse to elucidate pre-pregnancy and early gestation Er and Pr dynamics in a preeclamptic-like uterus. BPH/5 females display low circulating estrogen concentration during proestrus, followed by early gestation hypoestrogenemia, hyperprogesteronemia, and a spontaneous preeclamptic-like phenotype. Preceding pregnancy, the gene encoding Er alpha (Erα, *Esr1*) is upregulated in the diestrual BPH/5 uterus. At the peak of decidualization, *Esr1*, Er beta (Erβ, *Esr2*), and Pr isoform B (*Pr-B*) were upregulated in the BPH/5 maternal–fetal interface. At the protein level, BPH/5 females display higher percentage of decidual cells with nuclear Erα expression, as well as Pr downregulation in the decidua, luminal and glandular epithelium. In conclusion, we provide evidence of disrupted sex hormone signaling in the peri-conception period of preeclamptic-like pregnancies, potentially shedding some light onto the intricate role of sex hormone signaling at unexplored timepoints of human preeclampsia.

## 1. Introduction

Preeclampsia, a hypertensive disorder of pregnancy, is a leading cause of maternal and fetal morbidity and mortality [1]. In the United States, the incidence of preeclampsia has increased by 25% over approximately two decades, with a 6.7-fold increased risk of severe disease [2,3]. The syndrome is clinically characterized by new-onset hypertension after 20 weeks of gestation, coupled with at least one clinical sign of maternal hematopoietic, renal, hepatic, pulmonary, or neurological dysfunction [4,5,6]. A spectrum of pathophysiological pathways may lead to preeclampsia, with distinct upstream mechanisms ultimately converging on placental stress and release of inflammatory and antiangiogenic factors in maternal circulation [7,8,9]. In the mother, this process triggers systemic endothelial dysfunction and varied degrees of clinical disease. Correspondingly, preeclampsia subtypes have been gradually recognized, including early- (<34 weeks) and late-onset (>34 weeks) syndrome, each entailing unique risk factors, clinical presentations, and maternal–fetal complications [4,5,10,11]. Due to sustained placental damage and a relatively longer disease course, early-onset preeclampsia often leads to poorer outcomes compared to late-onset disease, including fetal growth restriction, and neonatal cardiorespiratory and neurological disorders [4].

As the understanding of the etiopathogenesis of preeclampsia deepens, the ripple effects arising from disturbances in the peri-conception period come into sharper focus [9,12,13]. Notably, recent studies have linked preexisting reproductive disorders, such as diminished ovarian follicular reserve and dysfunctional corpora lutea, to the development of preeclampsia [14,15,16]. Yet, the underlying mechanisms remain elusive. Following conception, the preclinical phase of early-onset preeclampsia is marked by perturbed trophoblast cell differentiation and invasion of the maternal decidua: a specialized endometrial layer with a key role in embryonic implantation and placental development [17]. Consequently, uterine vascular remodeling mediated by trophoblast cells is compromised, and uteroplacental malperfusion ensues [8,17]. Defective decidualization and perturbed endometrial-trophoblast crosstalk during early gestation are proposed as the genesis of impaired trophoblast cell dynamics [9,18]. However, the molecular underpinnings of those early preeclampsia events remain poorly understood.

Estrogen and progesterone govern pregnancy establishment and maintenance through activation of nuclear and membrane-bound receptors [19,20,21,22]. Fundamental roles of estrogen and progesterone signaling in the peri-conception period include induction of decidualization, angiogenesis, trophoblast cell differentiation, and uterine receptivity [22,23,24]. As pregnancy progresses, estrogen and progesterone modulate placental perfusion, local immune function, and maintain uterine quiescence [20,25]. Hence, it is not surprising that multiple studies have highlighted circulating estrogen and progesterone dysregulations in preeclamptic women during late gestation [20,26,27,28,29,30,31,32,33,34,35]. The expression of estrogen and progesterone receptors in the preeclamptic maternal–fetal interface has received less scientific attention, with conflicting findings reported [28,29,36,37]. Estrogen receptor alpha (ERα), estrogen receptor beta (ERβ), and progesterone receptor (PR) are classical nuclear receptors encountered in the cell cytoplasm and nucleus [37,38]. Upon ligand-receptor activation, hormone signaling is based on receptor dimerization and activation of response elements in the promoter region of target genes [37,38]. Notably, estrogen and progesterone are major drivers of uteroplacental ERα, ERβ, and PR gene and protein expression, while also promoting receptor degradation through proteasomal proteolysis [39]. Studies of estrogen and progesterone receptor expression and function in preeclamptic pregnancies are vastly limited to third trimester placental samples. Hence, such studies disregard endometrial molecular changes that may occur during early gestation, as well as the local crosstalk with the maternal placental bed [20,37]. Additionally, distinction between the subtypes of preeclampsia is often neglected, which may further hinder data interpretation [4,5,10,40].

With the aim to further understand pre-pregnancy and early gestation estrogen and progesterone receptor dynamics in a preeclamptic-like maternal–fetal interface, a series of studies were conducted with the Blood Pressure High Subline 5 (BPH/5) mouse model. Pregnant BPH/5 females recapitulate the main clinicopathological signs of human early-onset preeclampsia, including impaired placental vascular remodeling, late gestation superimposed hypertension, proteinuria, and fetal growth restriction [41,42,43,44]. Abnormal serum concentrations of 17β-estradiol and progesterone have been described in BPH/5 females pre-pregnancy and during early gestation [44,45]. This hormonal dysregulation is accompanied by delayed embryonic development, delayed decidualization, and impaired embryonic implantation [44]. Furthermore, the BPH/5 decidua is characterized by defective angiogenesis and exacerbated inflammation, features known to be modulated by sex steroid hormones [43,44,46,47]. Importantly, early gestation estrogen supplementation resulted in attenuation of uteroplacental molecular signaling and defective placentation in BPH/5 pregnancies [18]. In light of the intricate balance between sex steroid hormones and respective receptors, we hypothesized that uteroplacental Erα, Erβ, and Pr would also be dysregulated in the non-pregnant uterus and developing maternal–fetal interface of preeclamptic-like BPH/5 pregnancies.

## 2. Materials and Methods

### 2.1. Animal Husbandry

Experiments were performed using virgin BPH/5 and C57BL6 (C57) females from in-house colonies. The normotensive C57 control strain was used in the eight-way cross that originated the BPH/5 [1,2]. Adult (8–12 weeks of age) mice were housed in a climate-controlled environment (12 h light–dark cycle, 70.5–71 °F) and fed a standard chow diet (Purina 5001 rodent chow: 23% crude protein, 4.5% crude fat, 6% crude fiber, and 8% ash, Neenah, WI, USA) and ad libitum water. All animal procedures were approved by the Institutional Animal Care and Use Committees at Louisiana State University School of Veterinary Medicine and Pennington Biomedical Research Center and are in accordance with the PHS Guide for the Care and Use of Laboratory Animals. Vaginal cytology sampling was performed daily for at least two complete estrous cycles in virgin BPH/5 and C57 females, as previously described [48,49]. Cohorts of virgin females were then euthanized during proestrus or first day of cytological diestrus for pre-pregnancy investigations. Uterine wet weights were assessed immediately after euthanasia to corroborate vaginal cytology findings and were flash-frozen and cryopreserved at −80 °C until further analysis. Strain-matched timed matings were performed, with the day of detection of a vaginal copulatory plug designated as embryonic (e) 0.5. After copulation, singly housed BPH/5 and C57 females carrying natural pregnancies were euthanized during the peak of decidualization, at e7.5, which is a key timepoint in the etiopathogenesis of the BPH/5 preeclamptic-like phenotype [43,44,50]. Embryonic implantation sites and inter-implantation sites were harvested immediately after euthanasia, and immediately processed for molecular studies.

### 2.2. Quantitative (q) RT-PCR

Genomic DNA was eliminated, and total RNA was extracted from non-pregnant uteri and embryonic implantation sites using a commercial kit, according to the manufacturer’s instructions (Qiagen RNeasy, Hilden, Germany). The RNA ratios of absorbance and concentration were assessed using a NanoDrop Spectrophotometer (NanoDrop 200, ThermoFisher Scientific, Wilmington, DE, USA) and 1000 ng cDNA was synthetized using a commercial kit for reverse transcription (qScript cDNA, Quanta Biosciences, Gaithersburg, MD, USA). Quantification of gene expression levels was performed by qRT-PCR using SYBR Green (PerfeCTa SYBR Green FastMix, Quanta Biosciences, Gaithersburg, MD, USA). Each sample was run in triplicate, and mRNA expression was normalized to 18 s and analyzed using the ddCT method [51]. Sequence-specific amplification was confirmed by a single peak during the dissociation protocol following amplification, and by product size using gel electrophoresis.

Genes encoding the estrogen receptors Erα (*Esr1*), Erβ (*Esr2*), and Pr were assessed. When differential *Pr* expression was identified, relative abundances of specific *Pr* isoforms were calculated as fold changes of qRT-PCR cycle thresholds (Ct), in accordance with previous studies [25]. Three primer sets targeting different regions of the mouse *Pr* gene were used: (1) *Pr-A/B/C*, targeting the ligand-binding domain of *Pr*, shared by the three isoforms; (2) *Pr-A/B*, targeting the *Pr* gene segment shared by the isoforms *Pr-A* and *Pr-B*, but not *Pr-C*; (3) *Pr-B*, directed at a segment specific of the *Pr-B* isoform [25]. The dCT was obtained by mRNA expression normalization to the housekeeper gene (18 s), and the relative abundance (2-ddCT) of *Pr-A* and *Pr-C* was calculated after subtracting the abundance of *Pr-B* from *Pr-A/B*, and *Pr-A/B* from *Pr-A/B/C* [25]. Forward and reverse primer sequences are available on Appendix A.

### 2.3. Immunohistochemistry

Pregnant uteri were fixed using 4% paraformaldehyde for 24 h, followed by 70% ethanol (EtOH), and embedded in paraffin for immunohistochemistry assays. Paraffinized tissue blocks were cut at 5 µm using a microtome, mounted onto positively charged slides, and allowed to dry overnight at 35 °C. Mounted sections were deparaffinized in xylene and rehydrated using a graded ethanol series. Antigen retrieval was performed using sodium citrate buffer (10 mM, pH 6) solution at 90–95 °C. Endogenous peroxidase activity was quenched by incubating slides in 3% hydrogen peroxide in methanol for 30 min at room temperature (25 °C). Sections were rinsed with tris-buffered saline (TBS) containing 0.3% Triton X-100 (TBSt) for 5 min. Non-specific binding was prevented by incubating tissue sections in blocking buffer containing 2% (*v*/*v*) normal goat serum and 1% (wt/*v*) bovine serum albumin in TBSt for 1 h at room temperature. Tissues were then incubated in primary antibody for 1 h at room temperature, then overnight at 4 °C. Primary antibodies used were polyclonal rabbit anti-human Erα (1:500, PA1-309, Thermo Fisher Scientific, Braunschweig, Germany), and polyclonal rabbit anti-human Pr (1:400, AB63605, Abcam, Cambridge, MA, USA). Primary antibodies were omitted from negative control sections. After rinsing in TBSt, sections were incubated with secondary biotinylated goat anti-rabbit IgG (1:500) for 1 h at room temperature, followed by incubation in avidin-biotin complex at room temperature for 1 h (Vectastain Elite ABC Kit, Vector Laboratories, Burlingame, CA, USA). Diaminobenzidine solution was used to detect immunostaining (ImmPACT DAB Peroxidase Substrate, Vector Laboratories, Burlingame, CA, USA). Sections were counterstained with Mayer’s hematoxylin solution. Imaging was performed using the ZEISS Axioscan 7 slide scanner (Zeiss, Dublin, CA, USA). Quantification of immunostaining was performed using ImageJ software (NIH, version 1.54). A total of 4–7 embryonic implantation sites were analyzed. Tissue areas were randomly selected for immunostaining quantification by an investigator blinded to the study design. For each embryonic implantation site, a total of three areas were selected for decidua immunostaining quantification (50 µm × 50 µm). Likewise, three areas of undifferentiated sub-epithelial uterine stroma (50 × 50 µm), glandular epithelium (10 × 10 µm), and luminal epithelium (10 × 10 µm) from each inter-implantation site were selected for quantification. Standardization and color deconvolution of the selected areas were performed and the data for each tissue were archived, analyzed, and expressed as optical density. Additionally, four decidua areas (150 × 150 µm) from each embryonic implantation site were selected for assessment of decidual cell nuclear Erα and Pr immunostaining. Decidual cells were identified based on characteristic morphology [22]. All decidual cells in each area were included and classified as positive or negative for nuclear staining by an observer blinded to the study design. Experiments were performed in triplicate for each marker.

### 2.4. Statistical Analysis

Data analyses were performed using GraphPad Prism, version 10.02 (GraphPad Prism Software, Inc., La Jolla, CA, USA). Student’s *t*-tests were used for comparisons between BPH/5 and C57. One-way ANOVA and post hoc Tukey’s test were used for comparisons between multiple groups. Normality of residuals from the models were accessed and confirmed via Shapiro–Wilk tests and quantile–quantile (Q-Q) plots. Logarithmic (Log) transformation was performed for data normalization. Kruskal–Wallis’s test was performed for data that did not meet the normality criteria after Log transformation. Data are presented as mean ± SEM. Significance was set at *p* < 0.05.

## 3. Results

### 3.1. Uterine Esr1 Is Upregulated in Virgin BPH/5 Females during Diestrus

Given the abnormal circulating sex steroid hormone profile of BPH/5 females that precedes pregnancy, uterine expression of *Esr1*, *Esr2*, and *Pr* were first investigated in virgin adult BPH/5 females during proestrus and diestrus [45]. It has been previously reported that BPH/5 females display lower circulating 17β-estradiol concentration during proestrus [45]. Herein, uterine *Esr1*, *Esr2*, and *Pr* were not different between virgin proestrual BPH/5 and C57 females (Figure 1a–c, *p* > 0.05). Conversely, marked *Esr1* upregulation was noted in the BPH/5 non-pregnant uterus during diestrus, with relative expression approximately 10-fold higher than estrous-cycle-stage-matched C57 controls (Figure 1d, *p* = 0.017). Similar to proestrus, uterine *Esr2* and *Pr* were not different between virgin BPH/5 and C57 females during diestrus (Figure 1e,f, *p* > 0.05).

### 3.2. At the Peak of Decidualization, Esr1, Esr2, and Pr Isoform B Are Upregulated in the Preeclamptic-Like BPH/5 Maternal–Fetal Interface

During the peak of decidualization, at e7.5, *Esr1* relative expression was approximately 2-fold higher in implantation sites of BPH/5 females when compared to gestational stage matched C57 controls (Figure 2a, *p* = 0.006). While ERα is predominantly expressed in uterine tissue, ERβ expression is particularly high in the ovaries [22,38]. Accordingly, ERα seems to have a major role in uterine decidua development and function, while ERβ seems to have a lower to neutral participation in uterine estrogen-mediated responses [38]. In our studies, overall low *Esr2* mRNA abundance was noted in the embryonic implantation sites of both BPH/5 and C57 females. Nonetheless, *Esr2* upregulation was also seen in BPH/5 e7.5 embryonic implantation sites, with an approximately 2-fold higher relative expression when compared to C57 controls. (Figure 2b, *p* = 0.009).

Estrogen signaling, particularly through ERα, modulates endometrial PR expression [21]. During the peak of decidualization, *Pr* was approximately 1.5-fold higher in the BPH/5 maternal–fetal interface vs. C57 controls (Figure 2c, *p* = 0.029). At least three nuclear *PR* isoforms occur in the human and murine uterus and placenta, namely PR-A, PR-B, and PR-C, which are encoded by the same gene through different promoter activation [25,52]. Although the PR isoforms have distinct functions in specific tissues, the PR-B appears to be the main receptor associated with the well-established progesterone-mediated pathways in the non-pregnant uterus, while the PR-A and truncated PR-C may have inhibitory roles on PR-B [25]. Furthermore, contrasting pathways involved in trophoblast cell invasion seem to be modulated by specific progesterone binding to either PR-A or PR-B [52]. To further understand the *Pr* upregulation in the BPH/5 maternal–fetal interface, investigation of the distinct *Pr* isoforms was performed (Figure 2d–f). Interestingly, only *Pr-B* was upregulated in BPH/5 embryonic implantation sites at e7.5, with an approximately 1.5-fold higher relative mRNA expression in comparison to gestational age-matched C57 controls (Figure 2e, *p* = 0.027).

### 3.3. A Higher Population of Decidual Cells Display Nuclear Erα Expression in Early BPH/5 Pregnancies

Early in gestation, estrogen-primed endometrial stromal cells surrounding the embryos undergo progesterone-driven mesenchymal to epithelial transition to form the decidua, an indispensable tissue in embryonic implantation and placental development [53]. Spatial characterization of Erα was performed in the developing BPH/5 maternal–fetal interface at the peak of decidualization (Figure 3). At e7.5, marked Erα immunostaining was noted particularly in the outermost (i.e., secondary) decidual zone, as previously described in normal mouse pregnancies [22]. Interestingly, while the overall Erα mean immunostaining intensity was not different between BPH/5 and C57 decidual tissue (Figure 3a, *p* > 0.05), a higher percentage of BPH/5 decidual cells presented nuclear Erα immunostaining (Figure 3b, *p* < 0.0001). Positive cytoplasmic and nuclear Erα immunostaining was also observed in the glandular, luminal epithelium, and undifferentiated uterine stroma located between the inter-implantation site luminal epithelium and the myometrium, which was not different between hypertensive BPH/5 and normotensive C57 pregnancies (Figure 3c,d, *p* > 0.05).

### 3.4. At the Peak of Decidualization, Pr Is Downregulated in the BPH/5 Decidua, Luminal, and Glandular Uterine Epithelium

While *Pr* mRNA was upregulated in the e7.5 BPH/5 embryonic implantation sites, Pr protein expression downregulation was found at the cellular level (Figure 4). At e7.5, Pr immunostaining was particularly evident in the nuclei of decidual cells in both BPH/5 and C57 pregnancies, while mild Pr immunostaining was noted in the BPH/5 and C57 luminal and glandular epithelium. Interestingly, mean Pr staining intensity and the percentage of decidual cells with nuclear Pr staining were lower in BPH/5 females (Figure 4a,b, *p* < 0.05). Likewise, there was lower Pr immunostaining in the inter-implantation site luminal and glandular epithelium of BPH/5 females, while no difference was found between the BPH/5 and C57 sub-epithelial endometrial stroma (Figure 4c,d, US *p* < 0.05, GE and LE *p* < 0.001).

## 4. Discussion

This study demonstrates for the first time that estrogen and progesterone receptors are spatially dysregulated pre-pregnancy and during the peri-implantation maternal–fetal interface of a unique model of superimposed preeclampsia, the BPH/5 mouse. Multiple clinical studies suggest lower concentrations of estrogens in the serum and maternal–fetal interface of preeclamptic women during late gestation, albeit conflicting findings have been reported [20,26,27,28,29,30,31,32,33,54]. The role of progesterone in preeclampsia is a subject of even greater controversy, with clinical studies pointing to either increased, similar, or decreased circulating levels in late-gestation preeclamptic pregnancies [29,30,32,34,35]. Once preeclampsia is established, the associated vascular disorders seem to directly affect placental steroidogenesis, likely yielding the abnormal circulating estrogen and progesterone profiles seen in late gestation [31,32]. While subject to ongoing debate, placental sex steroid hormone receptor dysregulations have also been documented following preeclampsia [28,29,30,35]. Nevertheless, studies to date have only investigated estrogen and progesterone receptor expression in third trimester preeclamptic placentas. Hence, such investigations neglect the dynamic expression pattern of sex hormones and receptors throughout pregnancy, as well as the intricate paracrine signaling between the placenta and adjacent decidua [54].

Recent studies of women undergoing assisted reproductive procedures suggest that pre-conception reproductive endocrinopathies may have a role in the etiopathogenesis of preeclampsia [14,15,16]. Accordingly, early pregnancy events synergistically regulated by estrogen and progesterone seem to be particularly relevant in the development of the early-onset syndrome [9,24,55,56,57]. In a complex molecular crosstalk, estrogen and progesterone orchestrate uterine receptivity and decidualization [23,24,37]. In healthy pregnancies, uterine estrogen signaling promotes epithelial cell proliferation, vascular elongation, regulation of angiogenic factors, and immune modulation, creating a suitable environment for embryonic adhesion and implantation [38]. Conversely, progesterone inhibits estrogen-mediated epithelial cell proliferation, while concurrently promoting placental bed development by converting estrogen-primed endometrial stromal cells into specialized decidual cells [22,24]. The preclinical phase of early-onset preeclampsia is characterized by defective decidualization, decidual inflammation, and perturbed peri-conception endometrial-trophoblast crosstalk, culminating with defective trophoblast cell dynamics and inadequate uterine vascular remodeling [9,24,55,56,57]. Herein, we hypothesize that dysregulations of uterine estrogen and progesterone receptors in the peri-conception period have a key role in initiating the aforementioned cascade of events.

Unlike other preeclampsia animal models that require surgical or pharmacological intervention during mid-gestation, the BPH/5 preeclamptic-like phenotype occurs spontaneously [41,42,43,44,50,58]. Hence, this model provides a unique opportunity to investigate sex hormone receptor expression in a developing preeclamptic-like maternal–fetal interface. Pre-pregnancy, BPH/5 females present precocious pubertal onset, irregular estrous cyclicity and hypoestrogenism during proestrus [44,45,59]. Additionally, marked abnormalities characterize the BPH/5 peri-implantation period, including an apparent mismatch between embryonic and decidual development, and impaired embryonic implantation [44]. Interestingly, this is accompanied by a premature and depressed serum 17β-estradiol surge in the morning of e2.5, followed by lower 17β-estradiol concentration until the morning of e3.5 [44]. This was further associated with downstream aberrations in estrogen-dependent expression of leukemia inhibitory factor (Lif) in the BPH/5 implantation site. Concurrently, BPH/5 females display higher serum progesterone in the afternoon at e2.5 [44]. At e7.5, the BPH/5 decidua is characterized by angiogenic factor imbalance and marked inflammation, events tightly regulated by estrogen and progesterone [43,47,50,54]. Later in gestation, these abnormalities culminate with impaired uterine vascular remodeling, poor placental expansion, and a preeclamptic-like phenotype [41,42]. Further corroborating the role of estrogens in the preeclamptic-like BPH/5 phenotype, we have recently demonstrated that 17β-estradiol supplementation during early-gestation not only attenuates BPH/5 uteroplacental molecular profile, but also improves placental expansion towards the maternal decidua and fetal viability [18].

The nuclear ERα and ERβ are encoded by distinct genes in nonhomologous chromosomes, and the expression of both receptors is directly induced by estrogen [38]. In one study, ERα gene and protein expressions were lower in term placentae of apparently mild cases of preeclampsia compared to healthy pregnancies [37]. Conversely, at least two independent investigations have reported higher placental ERα in women with severe preeclampsia [36]. Following hypoestrogenemia in proestrus and early pregnancy, *Esr1* and *Esr2* are upregulated in the BPH/5 maternal–fetal interface, and higher decidual cell nuclear ERα immunostaining is observed during the peak of decidualization. Interestingly, the same apparent paradox of lower circulating estrogen and higher receptor expression has been reported in severe preeclamptic pregnancies at term [28]. A potential explanation is a compensatory *Esr1* upregulation in an estrogen deprived environment, as recently proposed in antiestrogen-responsive tumors [60]. The key role of ERα in the female reproductive tract is exemplified by uterine hypoplasia in *Esr1* knockout mice [38,61]. Even though ERβ is expressed in a wide variety of tissues, it does not seem to play a major role in the non-pregnant and early gestation uterus, as Erβ-null female mice do not display changes in uterine adenogenesis and function [38]. While both *Esr1* and *Esr2* were upregulated in the BPH/5 maternal–fetal interface, low *Esr2* mRNA abundance was noted. This is consistent with investigations in CD-1 mice carrying normal pregnancies, in which *Esr1* remained stable in the developing maternal–fetal interface from days 1–8 of pregnancy, while very low *Esr2* was detected [22]. Nevertheless, given the decline in human syncytiotrophoblast ERα expression, and the increase in ERβ expression throughout differentiation, the specific role of ERβ in the trophoblast cell phenotype warrants further investigation [62].

During early gestation, BPH/5 females not only display hyperprogesteronemia at e2.5, but also maternal–embryonic interface upregulation of *Pr-B* at e7.5. A contrasting profile was found at the protein level, with lower Pr expression in the decidua, glandular, and luminal epithelium. Uteroplacental estrogen signaling, particularly through activation of ERα, is a major driver of cell-specific PR expression [61]. Specifically, while estrogen decreases *Pr* expression in the mouse luminal epithelium, it increases *Pr* expression in the undifferentiated stroma and myometrium [22,61]. Hence, the BPH/5 spatial Pr expression is likely secondary to estrogen-Erα dysregulation. In addition to the well-described roles of progesterone in pregnancy establishment and maintenance, a paradoxical modulation of trophoblast cell invasion has been reported during the first trimester of human gestation [52]. Namely, progesterone-mediated activation of *PR-B* in early first trimester trophoblast cells lead to inhibition of invasion, potentially through inhibition of matrix metalloproteinase 2. Contrarily, trophoblast cell *PR-A* expression surpasses *PR-B* by the end of the first trimester, when progesterone signaling induces trophoblast cell invasion. Similar to human early-onset preeclampsia, impaired trophoblast cell invasion and shallow placentation precede the preeclamptic-like clinical signs in the BPH/5 mouse [42]. Altogether, those findings highlight the need of further investigations of progesterone-mediated inhibition of trophoblast cell invasion and placentation in preeclampsia.

In this study we have investigated the classical estrogen and progesterone nuclear receptors, which act as ligand-activated transcription factors at the promoter region of targeted genes [37,38]. Nevertheless, membrane-bound receptors with high estrogen or progesterone affinity have been described in multiple cell types, including the G protein-coupled estrogen receptor 1 (GPER), which mediates rapid, non-genomic responses to estrogen via secondary cellular pathways [33,63,64]. Recent studies have demonstrated *GPER* downregulation in term preeclamptic placentae, and GPER activation has been shown to induce trophoblast cell invasion [63,64,65]. Hence, GPER expression profile and function should also be considered in future investigations.

Although rodents are widely used as preclinical models of preeclampsia, important differences between mouse and human peri-implantation should not be overlooked [66,67,68]. Mice have a shallower, primarily interstitial, trophoblast cell invasion when compared to humans [69]. The mouse placenta has a labyrinthine interdigitation and is termed hemotrichorial, with three layers of trophoblast cells between maternal blood and fetal endothelial cells. Contrastingly, the human placenta has a villous maternal–fetal interdigitation, characterized as hemomonochorial due to the presence of a single trophoblast cell layer between the maternal blood and fetal capillaries [66]. Nevertheless, sex steroid hormone regulation of embryonic implantation and decidualization are highly conserved between the two species, including the rising levels of progesterone and estrogen in preparation for embryonic receptivity, and the estrogen surge preceding implantation and decidualization [70,71,72]. Hence, our findings with the BPH/5 mouse may offer valuable insights into estrogen and progesterone receptor dysregulations during timepoints that are particularly challenging to investigate in human pregnancies affected by preeclampsia.

## 5. Conclusions

In summary, dysregulation of uterine Erα, Erβ, and Pr precede the onset of preeclamptic-like pregnancy disorders in the BPH/5 mouse model. Herein we describe marked *Esr1* upregulation in the BPH/5 non-pregnant uterus during diestrus. Following the BPH/5 early gestation hypoestrogenemia and hyperprogesteronemia, marked *Esr1, Esr2,* and *Pr-B* upregulation occurs at the peak of decidualization, along with cell-specific Erα and Pr expression patterns. Given the key roles of uteroplacental estrogen and progesterone signaling on pregnancy establishment and development, the observed sex steroid hormone dysregulations before and during early pregnancy are likely to contribute to the development of preeclamptic-like disorders in this model. By providing evidence of disrupted sex hormone signaling in the peri-conception phase of preeclamptic-like BPH/5 pregnancies, this study may offer insights on potentially disrupted estrogen and progesterone signaling at unexplored timepoints of human preeclamptic pregnancies.

## Figures and Tables

**Figure 1 biology-13-00192-f001:**
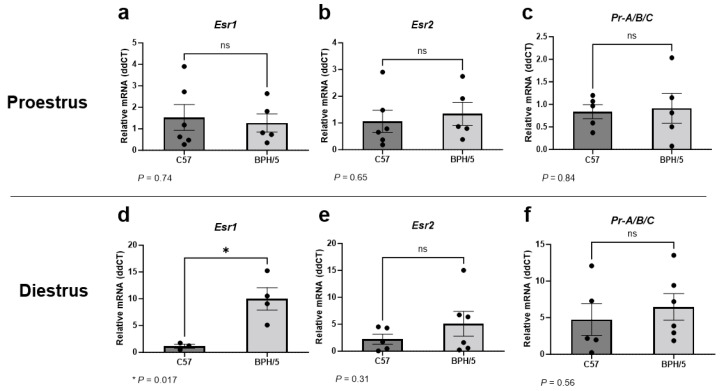
Uterine estrogen and progesterone receptor gene expression in the preeclamptic-like BPH/5 mouse prior to pregnancy. Relative gene expression of (**a**,**d**) estrogen receptor alpha (*Esr1*), (**b**,**e**) estrogen receptor beta (*Esr2*), and (**c**,**f**) progesterone receptor (*Pr-A/B/C*) in the uterus of virgin BPH/5 females and normotensive C57BL/6 (C57) during (**a**–**c**) proestrus and (**d**–**f**) diestrus. Student’s *t*-test, *n* = 3–6/group. Data expressed as mean ± SEM. * *p* < 0.05; ns = not significant.

**Figure 2 biology-13-00192-f002:**
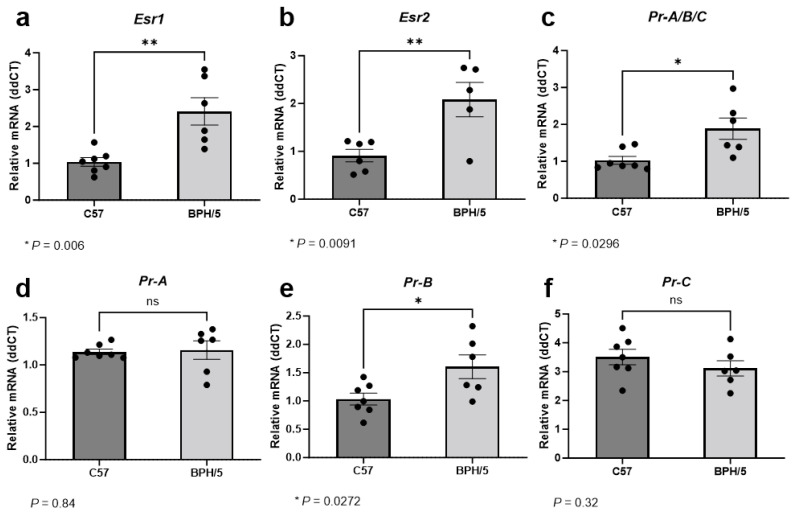
Estrogen and progesterone receptor gene expression in the preeclamptic-like BPH/5 maternal–fetal interface at the peak of decidualization. Relative gene expression of (**a**) estrogen receptor alpha (*Esr1*), (**b**) estrogen receptor beta (*Esr2*), and (**c**) progesterone receptor (*Pr*) in the embryonic implantation sites of BPH/5 females and normotensive C57BL/6 (C57) at embryonic day (**e**) 7.5. (**d**–**f**) Relative gene expression of progesterone receptor isoforms in the BPH/5 and C57 embryonic implantation sites at e7.5. Student’s *t*-test, *n* = 5–7/group. Data expressed as mean ± SEM. * *p* < 0.05; ** *p* < 0.01; ns = not significant.

**Figure 3 biology-13-00192-f003:**
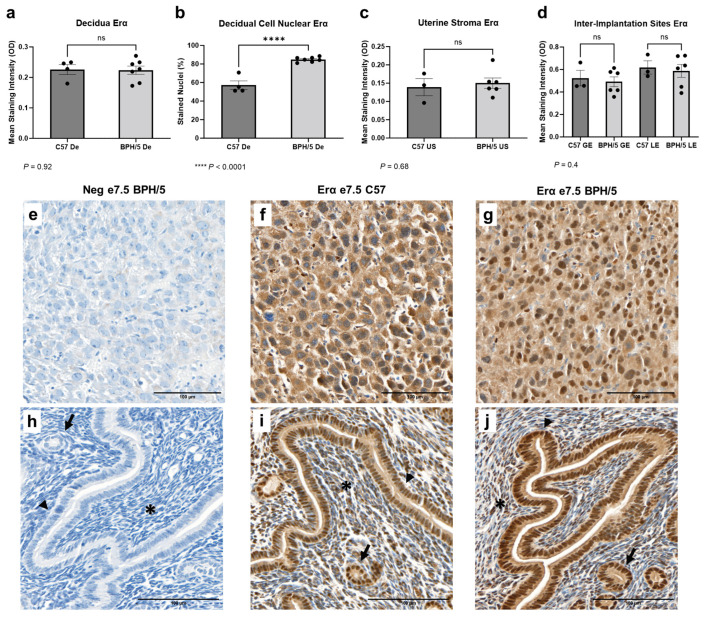
Estrogen receptor alpha (Erα) protein expression in the BPH/5 maternal–fetal interface at the peak of decidualization. (**a**) Mean Erα staining intensity in the BPH/5 and C57 decidua (De) at embryonic day (**e**) 7.5 and (**b**) percentage of decidual cells positive for nuclear Erα immunostaining; mean Erα staining intensity in (**c**) undifferentiated subepithelial uterine stroma (US); and (**d**) in the inter-implantation site glandular and luminal epithelia (GE and LE, respectively). (**a**–**c**) Student’s *t*-test, (**d**) one-way ANOVA. n = 4–7 embryonic implantation sites/group and n = 3–6 embryonic inter-implantation sites/group. Data expressed as mean ± SEM. **** *p* < 0.0001; ns = not significant. (**e**–**j**) Representative images of Er-α immunostaining in e7.5 BPH/5 and C57 (**e**–**g**) decidua and (**h**–**j**) inter-implantation sites as follows: (**e**,**h**) negative BPH/5 controls, (**f**,**i**) C57 Erα, and (**g**,**j**) BPH/5 Erα. Arrow = GE; arrow head = LE; asterisk = US. Scale bars: 100 µm.

**Figure 4 biology-13-00192-f004:**
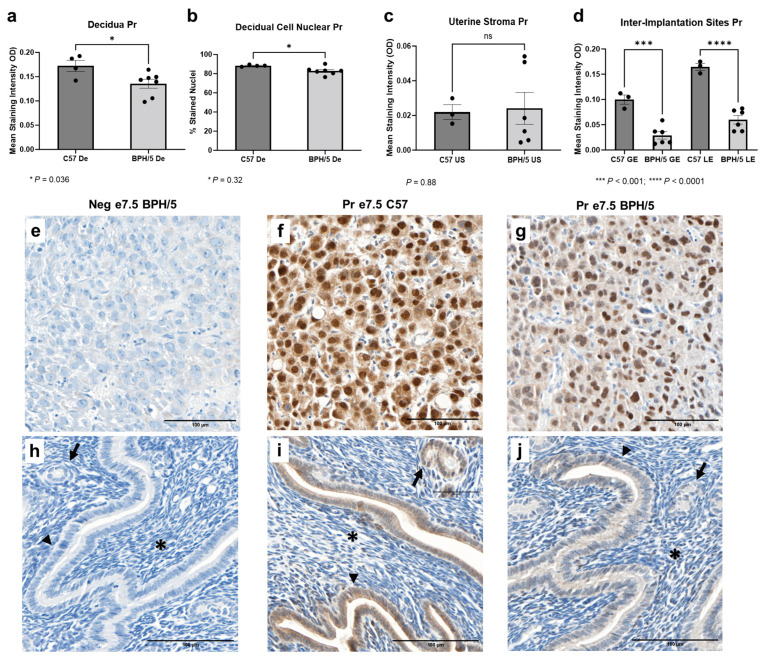
Progesterone receptor (Pr) protein expression in the BPH/5 maternal–fetal interface at the peak of decidualization. (**a**) Mean Pr staining intensity in the BPH/5 and C57 decidua (De) at embryonic day (**e**) 7.5, and (**b**) percentage of decidual cells positive for Pr nuclear immunostaining; mean Pr staining intensity in (**c**) undifferentiated subepithelial uterine stroma (US) and (**d**) inter-implantation site glandular and luminal epithelia (GE and LE, respectively). (**a**–**c**) Student’s *t*-test, (**d**) one-way ANOVA. n = 4–7 embryonic implantation sites/group and n = 3–6 embryonic inter-implantation sites/group. Data expressed as mean ± SEM. * *p* < 0.05; *** *p* < 0.001; **** *p* < 0.0001. (**e**–**j**) Representative images of Pr immunostaining in e7.5 BPH/5 and C57 (**e**–**g**) decidua and (**h**–**j**) inter-implantation sites as follows: (**e**,**h**) negative BPH/5 controls, (**f**,**i**) C57 Pr, and (**g**,**j**) BPH/5 Pr. Arrow = GE; arrow head = LE; asterisk = US. Scale bars: 100 µm.

## Data Availability

Data are contained within the article or Appendix A.

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
