# Peer review of "Estrogen and Progesterone Receptors Are Dysregulated at the BPH/5 Mouse Preeclamptic-Like Maternal–Fetal Interface"

_biology, 2024, doi:10.3390/biology13030192_

Round 1

Reviewer 1 Report

Comments and Suggestions for Authors

Dear Authors,

The investigation delves into a somewhat neglected domain by examining the expression of estrogen and progesterone receptors during the initial phases of gestation in a BPH/5 mouse model of preeclampsia. This focus is commendable as it addresses a crucial gap in preeclampsia research, shifting attention from the latter to the early stages of gestation – a critical period for the establishment of pregnancy and the potential origin of pathological changes.

The methodologies employed, including qRT-PCR and immunohistochemistry, are standard for this type of research. The application of appropriate statistical analyses (Student’s t-test, one-way ANOVA, post-hoc Tukey’s test) is duly noted. However, the limited sample size (n = 2-4/group for immunohistochemistry) may not provide sufficient power to detect small differences, a limitation that should be acknowledged. The observation of cell-specific receptor expression and dysregulation is intriguing. The use of the BPH/5 mouse model is appropriate for this study. However, one should be cautious when extrapolating these findings to human preeclampsia.

The discussion provides a solid overview of the potential implications of the findings and integrates them well within the existing body of literature. Yet, the study could benefit further from a more detailed discussion of how these findings advance the current understanding of preeclampsia’s etiopathogenesis, and a comparison between the model and the human condition would be beneficial.

The mention of membrane-bound receptors like GPER and their potential role is a strength. However, their inclusion in the study or future research directions could provide a more comprehensive understanding of hormone signaling in preeclampsia.

The introduction and discussion reflect a thorough understanding of the field. In the introduction, consider incorporating insights from the review from IJMS (MDPI) by Rybak-Krzyszkowska et al. to offer readers a foundational understanding of preeclampsia. Discussing the known molecular mechanisms, biomarkers, and the general landscape of preeclampsia research could provide a comprehensive backdrop against which your findings are set.

The study is conducted with commendable rigor, yet all research is inherently subject to biases. Reflecting on potential biases or limitations in the model, techniques, or interpretation of results would enhance the credibility of the study.

The authors hint at future research directions, but a more detailed roadmap for how this study’s findings could guide future research or clinical practice would be valuable.

The study provides important insights into the dysregulation of estrogen and progesterone receptors at the maternal-fetal interface in a BPH/5 mouse model of preeclampsia. The findings are valuable and contribute to a deeper understanding of the molecular events occurring during the early stages of preeclampsia. Future studies should aim to further elucidate the functional consequences of these receptor dysregulations and explore the potential of targeting these pathways for therapeutic intervention. The inclusion of a broader range of receptor types and a detailed comparative analysis with human data could significantly enhance the translational impact of the research.

In conclusion, the authors have done an excellent job. As a reviewer, I accept this manuscript with minor revisions.

Best regards!

Author Response

We are thankful for the constructive and comprehensive feedback provided. We have carefully considered the limitations raised, and have addressed each of them as follows:

“The limited sample size (n = 2-4/group for immunohistochemistry) may not provide sufficient power to detect small differences, a limitation that should be acknowledged”. The authors are grateful for the comment. Further immunostaining quantification was performed, and the statistics were re-analyzed. A total 4-7 embryonic implantation sites and 3-6 inter implantation sites are now included in the immunohistochemical analyses. Furthermore, the statistical analyses were revised to reflect our experimental units, as suggested by reviewer 2.

The use of the BPH/5 mouse model is appropriate for this study. However, one should be cautious when extrapolating these findings to human preeclampsia” and “Yet, the study could benefit further from a more detailed discussion on how these findings advance the current understanding of preeclampsia’s etiopathogenesis and a comparison between the model and the human condition would be beneficial”:  The authors fully agree with those statements. While this pre-clinical animal model provides a unique opportunity to investigate early pregnancy events in the context of preeclampsia, such results should be carefully considered when translated to humans. Additional comments in this regard were included in the discussion (lines 426-440).

“The mention of membrane-bound receptors like GPER and their potential role is a strength. However, their inclusion in the study or future research directions could provide a more comprehensive understanding of hormone signaling in preeclampsia.” We are in agreement with the need of such future investigation, as emphasized in the discussion lines 417-425.

“The introduction and discussion reflect a thorough understanding of the field. In the introduction, consider incorporating insights from the review from IJMS (MDPI) by Rybak-Krzyszkowska et al. to offer readers a foundational understanding of preeclampsia.” The proposed literature review was incorporated in the manuscript introduction.

Reviewer 2 Report

Comments and Suggestions for Authors

In this manuscript, the authors tested the hypothesis that estrogen and progesterone receptor aberrant expressions occur to the maternal-fetal interface during early gestation. Taking the advantage of a spontaneous model of superimposed preeclampsia, the Blood Pressure High Subline 5 (BPH/5) mouse, the authors performed qPCR and IHC staining to compare the expression levels of estrogen and progesterone receptors between the maternal-fetal interfaces of preeclamptic and normal pregnant C57BL6 female mice. The findings of this manuscript indicate the potential involvement of sex steroid hormone signaling in human preeclamptic pregnancies.

Generally speaking, this is a straightforward study, and an important addition to the literature. My comments are listed as below:

1) Are there any differences for the basal level expressions of estrogen and progesterone receptors in the uterine epithelium between the non-pregnant female mice of BPH/5 and C57BL6?

2) In Fig. 2A, for the quantification of staining intensity, I assume that each dot stands for each picture. The author should calculate the mean value of the staining intensity for each mouse, then make each dot represents one mouse. Additionally, please add more n, because if n = 2/group, the group size is too small.

3) The suggestion for Fig. 2A is also applied to Fig. 3A.

4) The authors need to improve the resolution of staining pictures in Fig. 2d and Fig. 3d.

Author Response

We are thankful for the feedback provided. The specific comments were addressed, as outlined in this response to the reviewer, and have certainly contributed to the overall manuscript quality.

1) The authors are thankful for the question and acknowledge the importance of pre-pregnancy investigations in this spontaneous model of preeclampsia. To elucidate the BPH/5 mouse baseline uterine expression of estrogen and progesterone receptors, we have performed Esr1, Esr2, and Pr quantitative real-time PCRs in uterine samples from BPH/5 and C57 females during both proestrus and diestrus, as further detailed in the revised manuscript. Interestingly, while Esr2 and Pr were not different in the uterus of BPH/5 and C57 females during proestrus and diestrus, Esr1 was approximately 10-fold higher in BPH/5 uteri during non-pregnant diestrus.

2 and 3) We genuinely appreciate those comments. Additional immunostaining quantification was performed to increase our sample size. Moreover, all immunohistochemistry statistics and graphs were updated to reflect our experimental units (embryonic implantation and inter-implantation sites), which is consistent with numerous investigations from our group and others (e.g, PMID: 27159542, PMID: 31780662).

4) The authors are thankful for the comment. The immunohistochemistry pictures were updated to improve quality.